# Acute Myocardial Infarction, Subclavian Vein Thrombosis, and Pulmonary Embolism Secondary to COVID-19—A Case Report

**DOI:** 10.3390/medicina59040656

**Published:** 2023-03-26

**Authors:** Nevena Georgieva Ivanova

**Affiliations:** 1Department of Urology and General Medicine, Faculty of Medicine, Medical University of Plovdiv, 4000 Plovdiv, Bulgaria; nevenai@yahoo.com; Tel.: +35-98-8913-0416; 2St. Karidad MBAL, Karidad Medical Health Center, Cardiology, 4004 Plovdiv, Bulgaria

**Keywords:** COVID-19, thrombotic complications, myocardial infarction, subclavian vein thrombosis

## Abstract

*Introduction*: Coronavirus disease 2019 (COVID-19) is an infectious disease caused by the severe acute respiratory syndrome coronavirus 2 (SARS-CoV-2). The majority of infected patients develop the clinical picture of a respiratory disease, although some may develop various complications, such as arterial or venous thrombosis. The clinical case presented herein is a rare example of sequential development and combination of acute myocardial infarction, subclavian vein thrombosis (Paget Schroetter syndrome), and pulmonary embolism in the same patient after COVID-19. *Case presentation*: A 57-year-old man with a 10-day history of a SARS-CoV-2 infection was hospitalized with a clinical, electrocardiographic, and laboratory constellation of an acute inferior-lateral myocardial infarction. He was treated invasively and had one stent implanted. Three days after implantation, the patient developed shortness of breath and palpitation on the background of a swollen and painful right hand. The signs of acute right-sided heart strain observed on the electrocardiogram and the elevated D-dimer levels strongly suggested pulmonary embolism. A Doppler ultrasound and invasive evaluation demonstrated thrombosis of the right subclavian vein. The patient was administered pharmacomechanical and systemic thrombolysis and heparin infusion. Revascularization was achieved 24 h later via successful balloon dilatation of the occluded vessel. *Conclusion*: Thrombotic complications of COVID-19 can develop in a significant proportion of patients. Concomitant manifestation of these complications in the same patient is extremely rare, presenting at the same time, quite a therapeutic challenge to clinicians due to the need for invasive techniques and simultaneous administration of dual antiaggregant therapy combined with an anticoagulant treatment. Such a combined treatment increases the hemorrhagic risk and requires a serious accumulation of data for the purpose of a long-term antithrombotic prophylaxis in patients with such pathology.

## 1. Introduction

Coronavirus disease 2019 (COVID-19) is an infectious disease caused by the severe acute respiratory syndrome coronavirus 2 (SARS-CoV-2). Most of the affected patients develop the clinical picture of a respiratory disorder involving the upper or lower respiratory tracts [1]. The growing body of data from research undertaken around the world suggests that the presentation of the disease or its later complications may be of vascular origin in certain individuals who develop arterial or venous thrombosis [2]. The clinical case we present here is a rare example of the sequential development of the combination of acute myocardial infarction, subclavian vein thrombosis (Paget Schroetter syndrome), and pulmonary embolism in the same patient after COVID-19, which required interventional procedures to diagnose and treat.

## 2. Case Presentation

A 57-year-old man was admitted to the Cardiology Department because of chest discomfort with vegetative symptoms in a typical risk constellation of accompanying arterial hypertension, atherogenic dyslipidemia, and smoking. He reported a SARS-CoV-2 infection 10 days prior to admission, which he treated at home. On physical examination, the patient appeared in an impaired general condition, with signs of pulmonary congestion (bilaterally at the bases). The electrocardiogram demonstrated an acute inferior-lateral myocardial infarction, which was also confirmed on the laboratory tests—increased levels of creatine kinase 270 U/L (normal < 170 U/L) and troponin 0.1 ng/mL (normal < 0.01 ng/mL). An invasive treatment strategy was implemented performing a percutaneous coronary intervention (PCI) using right radial access. The infarct-related artery, in this case the right coronary artery, was treated with implantation of one drug-eluting stent (DES) in the distal segment, and high-pressure post-dilatation was performed immediately after. An excellent angiographic result was achieved, and the blood flow was restored, as assessed with the thrombolysis considering the myocardial infarction 3 (TIMI 3) score. TIMI is widely used to assess coronary blood flow in acute coronary syndromes during percutaneous angioplasty. Grades range from 0–3 (0, no perfusion; 3, complete perfusion). Drug therapy included the administration of intravenous and oral medications—nitroglycerin, unfractionated heparin, antiplatelet agents (ticagrelor and acetylsalicylic acid), a statin (rosuvastatin), and a metabolic agent (trimetazidine dihyrochloride). The patient was discharged in good general condition and without any complaints. Three days following the intervention, edema and soreness were observed on the left hand, and after another two days, the patient was admitted into the ward again as an emergency due to shortness of breath, palpitations, and chest heaviness. The electrocardiogram indicated acute right-sided heart strain (right axis deviation), deep S wave in the first standard lead, Q and negative T waves in the third standard lead (with a S1Q3T3 pattern), ST depression and negative T waves in the fifth and sixth precordial leads (V5–V6), and pathological Q waves in the second and third standard leads and in the limbs (II, III, aVF). Laboratory studies showed elevated levels of D-dimer (>0.5/+/positive) and negative markers of myocardial necrosis (troponin 0.01 ng/mL). These deviations from normal values steered the diagnostic reasoning towards acute pulmonary embolism because of a suspected deep vein thrombosis of the arm. The Doppler ultrasound study confirmed the presence of a thrombotic mass in the right subclavian vein (Figure 1A). The diagnostic-therapeutic process continued with an emergency invasive examination and access from the right median cubital vein. Phlebography revealed a massive thrombotic occlusion of the subclavian vein from its transition from the axillary vein to the point it drains into the superior vena cava (Figure 1B) and a mosaic blood flow and multiple polycyclic shadows (Figure 1C). A developed collateral network was observed suprascapularly with outflow of the contrast material to the superior vena cava, with complete late contrast of the latter (Figure 1D).

The procedure continued with the use of a guide wire that passed through the occlusion via a microcatheter that reached the right atrium. Several guidewires were changed subsequently until at the end, a pharmacomechanical thrombolysis catheter was placed, through which we could inject therapeutic agents directly into the superior vena cava, while the occlusion was filled through the side holes (Figure 2A). The treatment continued with systemic fibrinolysis and a 24-h heparin infusion under strict control and dosing according to the results of the coagulogram. A control phlebography was performed the next day to assess the results. The whole right subclavian vein was clearly outlined now from the beginning to the end, with some thrombotic masses and residual stenosis still observed in the lumen due to organized fibrosis distally; most collaterals were missing because of centralization of the blood flow and good blood flow through the vessel (Figure 2B). The invasive treatment continued by placing a 6-mm embolic protection device to conduct 4-mm balloon dilatation. Dilatation ruptured the fibrotic masses and plaques (Figure 2C) while thromboaspiration evacuated the organized blood clots.

These procedures finally resulted in recanalization of the right subclavian vein with direct outflow to the superior vena cava, which was stained using the contrast material. The patient was discharged from hospital in a good general condition, without shortness of breath. The dilated, previously visible venous vessels on the right arm disappeared. For the at-home treatment, the following drugs were prescribed to the patient: pentasaccharide approved for the treatment of pulmonary embolism (fondaparinux 7.5 mg once daily administered subcutaneously according to body weight 50–100 kg) for 21 days, followed by a non-vitamin K oral anticoagulant (rivaroxaban 20 mg daily), an antiplatelet agent (clopidogrel 75 mg daily), acetylsalicylic acid (75 mg daily for 3 months), nitrate (molsidomine 2 mg twice daily), a beta-blocker (nebivolol 5 mg daily), angiotensin-converting enzyme inhibitor (ramipril 5 mg daily), proteolytic enzyme product (fibrozyme 320 mg 3 times daily), flavonoid medication (hydrosmin 200 mg 3 times a day), and antibiotics (ciprofloxacin 500 mg 2 times a day for 7 days and amoxicillin/clavulanic acid 1000 mg a day for 7 days). The long-term anticoagulant therapy combined with dual antiplatelet therapy is expected to have a positive effect on the patient to achieve complete recanalization of the right subclavian vein.

## 3. Discussion

Thrombotic complications in COVID-19 are of crucial importance since they have a direct bearing on the disorder’s prognosis and outcome, affecting them adversely [3]. The exact mechanism that causes these life-threatening conditions is still unknown. It could quite possibly be either excessive inflammation, triggered platelet aggregation, activation of some coagulation factors, or intravascular coagulation. A correlation has been found between the elevated levels of D-dimer, prothrombin, interleukin-6, and fibrinogen [4]. Another mechanism that could be hypothetically involved is endothelial inflammation and dysfunction. Vascular endothelial damage is probably caused by the direct impact of the virus entering the cell through overexpression of the angiotensin-converting enzyme receptor 2 (ACE 2) [5,6]. This results in the loss of the receptor’s anticoagulant function, which, combined with the release of other procoagulant factors such as the von Willebrand factor, leads to a stronger hypercoagulative response of the body. In addition, binding to ACE 2 is associated with an elevated concentration of angiotensin II, which exerts inflammatory and pro-thrombotic effects [7]. All this results in subsequent microvascular thromboses and complications [8,9].

The incidence of thrombotic complications in patients infected with SARS-CoV-2 ranges from 7.7% to 49%, which is much higher than that in patients without COVID-19, with venous thromboses outnumbering arterial thromboses [10]. Concomitant factors such as older age, male sex, an underlying coronary artery disease, and raised D-dimer levels increase the risk for thrombotic events according to an observational study on 3334 patients hospitalized for COVID-19. Elevated D-dimer levels might be a surrogate marker of the disease severity in patients with critical forms of the disease admitted for treatment in intensive care units [11]. A study on 1099 patients with COVID-19 in China reported D-dimer levels equal to or higher than 0.5 mg/L in 46% of the cases. Furthermore, SARS-CoV-2 infection is associated with a variety of laboratory changes, including hematological, immunological, biochemical, and hormonal changes. Significant changes were detected in the fibrin degradation products and ferritin, which were raised, as well as the prothrombin time and activated partial thromboplastin time, which were both prolonged. Lower fibrinogen and antithrombin levels were reported in patients who died of COVID-19. Common coagulation activation, dysregulated thrombin production, and defective natural anticoagulants or hyperfibrinolysis are the most likely underlying pathophysiological mechanisms responsible for these laboratory abnormalities [12].

A study has found that critically ill patients have an 8% risk of developing acute myocardial injury, with a 13-fold increase in the incidence [13]. In an Italian study on 28 patients with confirmed SARS-CoV-2 infection, ST-elevation myocardial infarction has been reported to represent the first clinical manifestation of COVID-19 in 85.7% of these cases [14]. The incidence of pulmonary embolism has been reported in the literature to be between 2.6% and 8.9% in hospitalized patients with COVID-19, occurring in as much as one-third of the patients requiring admission to an intensive care unit, despite the administered standard prophylactic anticoagulation [15]. An interesting retrospective, single-center study on 190 hospitalized patients with SARS-CoV-2 was conducted to describe 12- leads ECG alterations and to point out their prognostic value. The study reported that right ventricular strain was associated with a higher 28-day mortality risk [16].

Paget Schroetter’s syndrome (PSS) is a thrombosis of the axillary and subclavian veins; it is associated with some anatomical features of the course of these veins—proximity to the clavicle, the first rib, the anterior scalene, and the subclavian muscles. James Paget described it first in detail in 1875, and von Schroetter in 1884 put forward the hypothesis that this syndrome occurs as a result of repetitive musculoskeletal impact during exercise, when vascular compression becomes such that it leads to stasis, thrombosis, and impaired drainage. PSS accounts for 10–20% of all venous thromboses of an upper extremity, and approximately 1–4% of venous thromboses in general, making it relatively rare [17,18]. Single cases of its manifestation in patients with COVID-19 have been published in the literature. We found no information about the combination of PSS with acute myocardial infarction and pulmonary embolism in the same patient secondary to COVID-19, which makes the presented clinical case extremely valuable for practice.

## 4. Conclusions

Arterial and venous thromboses and thromboembolic events occurring during and after SARS-CoV-2 infection in a significant number of individuals are considered very serious complications Although their co-occurrence in the same patient is highly rare, it is a therapeutic challenge due to the need to use invasive techniques and simultaneous administration of dual antiaggregant treatment (acetylsalicylic acid and clopidogrel) in combination with an anticoagulant (rivaroxaban). This increases the risk of bleeding and requires the accumulation of data on long-term antithrombotic prophylaxis in patients with such pathology.

## Figures and Tables

**Figure 1 medicina-59-00656-f001:**
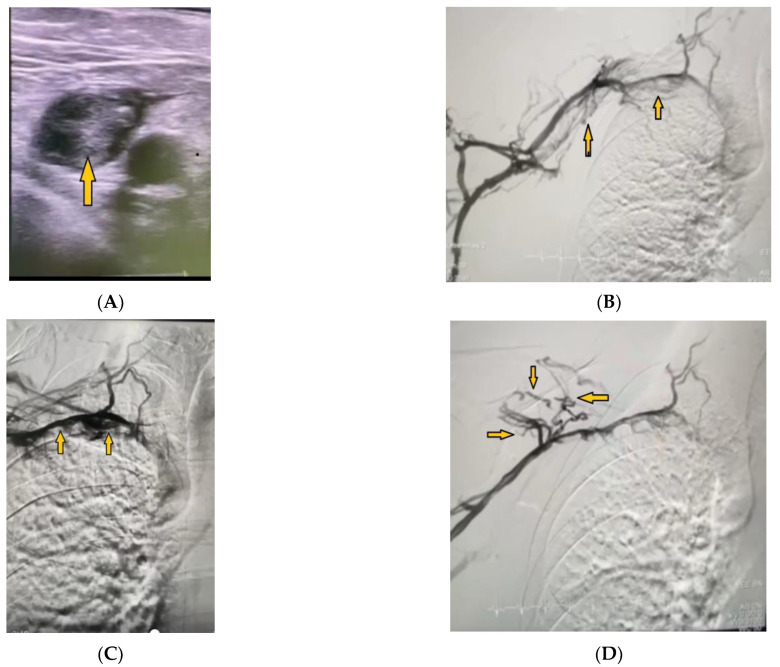
(**A**) Doppler ultrasound revealed subclavian vein thrombosis (yellow arrow). (**B**) Phlebography showed massive thrombotic occlusion of the right subclavian vein (yellow arrows). (**C**) Phlebography revealed subclavian vein thrombosis, mosaic blood flow, and multiple polycyclic shadows (yellow arrows). (**D**) Suprascapular collateral vascular network (yellow arrows).

**Figure 2 medicina-59-00656-f002:**
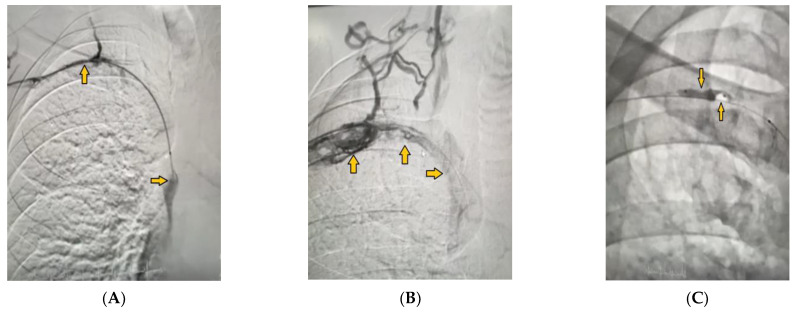
(**A**) Pharmaco-mechanical thrombolysis was performed using a catheter placed in the superior vena cava (yellow arrow) and subclavian vein (yellow arrow). (**B**) The outlined subclavian vein with residual thrombotic masses and stenosis due to organized fibrosis (yellow arrows). (**C**) Subclavian vein balloon dilatation causing the rupture of fibrotic masses (yellow arrows).

## Data Availability

The data presented in this study are available on request from the corresponding author.

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
