# Peer review of "Acute Myocardial Infarction, Subclavian Vein Thrombosis, and Pulmonary Embolism Secondary to COVID-19—A Case Report"

_medicina, 2023, doi:10.3390/medicina59040656_

Round 1
Reviewer 1 Report
The article is a case presentation of a patient with multiple thromboembolic complications secondary to COVID-19. The manuscript is well-written and the imaging data very illustrative.
Minor corrections must be done:
- "COVID-19 infection" is incorrect. Please write either "COVID-19" OR "SARS-CoV-2 infection"
- the format of the references must be corrected according to Instructions for authors
- In the Discussion section, the author must add some data about the alterations of coagulation tests in COVID-19. The following references can be added:
1. Rasmi Y, Saavedra LPJ, Cozma MA, El-Nashar H, Aly S, Fahmy N, Eldahshan O, El-Shazly M, Dobrica EC, Kord-Varkaneh H, Diaconu CC, Gaman MA. Laboratory findings in COVID-19 – alterations of hematological, immunological, biochemical, hormonal and other lab panels: a narrative review. Journal of Mind and Medical Sciences. 2022;9(1):38-55. DOI: 10.22543/7674.91.P3855.
2. Diaconu CC. Thromboembolism in patients with COVID-19. Arch Balk Med Union. 2021;56(3):281-283.DOI 10.31688/ABMU.2021.56.2.139
3. Diaconu CC. Cardiovascular complications of COVID-19. Archives of the Balkan Medical Union. 2021;56(2):139-141. DOI 10.31688/ABMU.2021.56.2.139
Reviewer 2 Report
The paper entitled “Acute Myocardial infarction, subclavian vein thrombosis and pulmonary embolism secondary to COVID-19 infection” well describe a clinical case that highlight the very frequent complications of SARS-CoV-2 infection: the thrombotic events.
The paper has a good scientific soundness and focused the importance of diagnosis and antithrombotic therapy.
In order to improve the quality of presented work, I would to suggest:
1. Correct some punctual error:
a. In line 1 delete the word “Title”
b. In line 38 delete the first open round bracket and the words “a virus,”
c. In line 174 change “Thrombotic (arterial and venous)” with “Arterial and venous thrombosis”
2. In the case presentation is not correct the description of “hemodynamic instability with hypotension (arterial pressure 110/70 mmHg). Please add more detail (mean arterial pressure, lactate, diuresis) or delete the sentence about arterial pressure.
3. You should report the D-Dimer value at first admission to hospital. Are you sure that hemodynamic instability was given by a distal stenosis in the right coronary artery? Or there was yet a pulmonary embolism?
4. Please describe the vascular access during PCI (right radial access?).
5. In line 72-26 are described the ECG findings. It’s important to report the prognostic value of such features. I suggest to cite “Prognostic Value of 12-Leads Electrocardiogram at Emergency Department in Hospitalized Patients with Coronavirus Disease-19. Savelloni G, Gatto MC, Cancelli F, Barbetti A, Cogliati Dezza F, Franchi C, Carnevalini M, Galardo G, Bucci T, Alessandroni M, Pugliese F, Mastroianni CM, Oliva A. J Clin Med. 2022 Apr 30;11(9):2537. doi: 10.3390/jcm11092537.”
6. In order to rationalize the figures, I suggest to group the first 4 figure in one panel (Figure 1 a, b, c, d) and the last three figures in another one panel (Figure 2 a, b, c).
7. At line 126 it is described the at-home treatment with rivaroxaban 20 mg daily. The pharmacological data sheet of rivaroxaban indicates that the therapy for pulmonary embolism is 15 mg of rivaroxaban twice a day for 21 days followed by rivaroxaban 20 mg (or 15 mg) once a day. Please add the information about the first 20 days of anticoagulant treatment. Low weight molecular heparin was used? What about the dosage?
8. The clinical presentation justifies the high bleeding risk of triple antithrombotic therapy. For how many times is prescribed?
9. In the discussion is described the possible mechanism of SARS-CoV-2 vascular damage. Please add in the reference “Bradyarrhythmias in patients with SARS-CoV-2 infection: A narrative review and a clinical report. Gatto MC, Persi A, Tung M, Masi R, Canitano S, Kol A. Pacing Clin Electrophysiol. 2021 Sep;44(9):1607-1615. doi: 10.1111/pace.14308. Epub 2021 Jul 14
In my humble opinion the case is of great interest for scientific community and will be considerable for publication after minor revisions.
Author Response
Please see the attachement
